# Metformin Therapy Effects on the Expression of Sodium-Glucose Cotransporter 2, Leptin, and SIRT6 Levels in Pericoronary Fat Excised from Pre-Diabetic Patients with Acute Myocardial Infarction

**DOI:** 10.3390/biomedicines9080904

**Published:** 2021-07-28

**Authors:** Celestino Sardu, Nunzia D’Onofrio, Michele Torella, Michele Portoghese, Simone Mureddu, Francesco Loreni, Franca Ferraraccio, Iacopo Panarese, Maria Consiglia Trotta, Gianluca Gatta, Marilena Galdiero, Ferdinando Carlo Sasso, Michele D’Amico, Marisa De Feo, Maria Luisa Balestrieri, Giuseppe Paolisso, Raffaele Marfella

**Affiliations:** 1Department of Advanced Medical and Surgical Sciences, University of Campania “Luigi Vanvitelli”, 80138 Naples, Italy; ferdinandocarlo.sasso@unicampania.it (F.C.S.); giuseppe.paolisso@unicampania.it (G.P.); raffaele.marfella@unicampania.it (R.M.); 2Department of Precision Medicine, University of Campania “Luigi Vanvitelli”, 80138 Naples, Italy; nunzia.donofrio@unicampania.it (N.D.); marialuisa.balestrieri@unicampania.it (M.L.B.); 3Department of Translational Medical Sciences, University of Campania “Luigi Vanvitelli”, 80138 Naples, Italy; michele.torella@unicampania.it (M.T.); francesco.loreni1993@gmail.com (F.L.); franca.ferraraccio@unicampania.it (F.F.); iacopo.panarese@gmail.com (I.P.); marisa.defeo@unicampania.it (M.D.F.); 4Department of Cardiac Surgery, Santissima Annunziata Hospital, 07100 Sassari, Italy; michele.portoghese@aousassari.it (M.P.); celestino.sardu@unicamillus.it (S.M.); 5Department of Experimental Medicine, University of Campania “Luigi Vanvitelli”, 80138 Naples, Italy; mariaconsiglia.trotta2@unicampania.it (M.C.T.); gianluca.gatta@unicampania.it (G.G.); marilena.galdiero@unicampania.it (M.G.); michele.damico@unicampania.it (M.D.); 6Mediterranea Cardiocentro, 80122 Naples, Italy

**Keywords:** pre-diabetes, acute myocardial infarction, metformin, pericoronary fat, sodium-glucose cotransporter 2 protein, leptin

## Abstract

Background and purpose: pericoronary fat over-inflammation might lead to the development and destabilization of coronary plaque in patients with pre-diabetes (PDM). Notably, pericoronary fat could over-express the sodium-glucose cotransporter 2 (SGLT2) and leptin, along with decreased sirtuin 6 (SIRT6) expression in PDM vs. normoglycemic (NG) patients undergoing coronary artery bypass grafting (CABG) for acute myocardial infarction (AMI). However, in the current study, we evaluated inflammatory markers, SGLT2, SIRT6, and leptin levels in pericoronary fat and, subsequently, 12-month prognosis comparing PDM to NG subjected to CABG for AMI. In addition, we evaluated in PDM patients the effects of metformin therapy on SIRT6 expression, leptin, and SGLT2 levels, and assessed its beneficial effect on nitrotyrosine and inflammatory cytokine levels. Methods: we studied AMI patients referred for CABG, divided into PDM and NG-patients. PDM patients were divided into never-metformin users and metformin users. Finally, we evaluated major adverse cardiac events (MACE) at a 12-month follow-up. Results: the MACE was 9.1% in all PDM and 3% in NG patients (*p* < 0.05). Metformin users presented a significantly lower MACE rate in PDM than never-metformin users (*p* < 0.05). PDM showed higher inflammatory cytokines, 3-nitrotyrosine levels, SGLT2, and leptin content, and decreased SIRT6 protein levels in pericoronary fat compared to NG-patients (*p* < 0.05). PDM never-metformin-users showed higher SGLT2 and leptin levels in pericoronary fat than current-metformin-users (*p* < 0.05). Conclusions: metformin therapy might ameliorate cardiovascular outcomes by reducing inflammatory parameters, SGLT2, and leptin levels, and finally improving SIRT6 levels in AMI-PDM patients treated with CABG.

## 1. Introduction

The sodium-glucose cotransporter 2 (SGLT2) is a protein increased by over-inflammation in patients with type 2 diabetes mellitus and implied in the process of atherosclerotic plaque destabilization and rupture [1]. These processes could consequently lead to worse clinical outcomes [2]. Moreover, the therapeutic inhibition of SGLT2 could reduce these adverse events and ameliorate clinical outcomes in treated patients [1]. Intriguingly, this could be due to, on the one hand, anti-hyperglycemic effects [1], and on the other hand, non-anti-hyperglycemic effects [3]. Indeed, SLGT2 inhibition modulates the leptin expression and reduces the visceral fat activity and the inflammatory burden [3].

Interestingly, the over-activity and inflammation of the visceral fat have also been evidenced in patients with pre-diabetes mellitus (PDM) [4]. Notably, first, this last effect could be mainly evidenced in PDM patients with a decreased sirtuin 6 (SIRT6) expression at the adipose tissue level [5]. Secondly, PDM affects more than 38% of people in the adult population and shows an increased risk of cardiovascular disease and a higher rate of major adverse cardiac events (MACE) [4]. On the other hand, fewer data have been reported about the expression of SGLT2 protein in the ectopic fat of PDM patients. In addition, there are no conclusive data about the expression of SGLT2 protein and its association with leptin over-activity in ectopic fat of PDM patients. Finally, but not less relevant, less is reported about the therapeutic inhibition of the SLGT2 axis and its effects on clinical outcomes. Therefore, in the current study, we might speculate that, in PDM patients, the coronary endothelial dysfunction (and the worse clinical outcomes) might be caused by over-inflammation due to increased adipose tissue deposits. Indeed, the coronary arteries are the most atherosclerosis-prone arteries with abundant adipose tissue surrounding them [6]. Notably, the pericoronary fat is an active component of coronary walls and is involved in atherosclerosis progression via over-inflammation [6,7,8].

Intriguingly, in PDM patients, metformin, which is a hypoglycemic drug, reduces the inflammatory tone and leptin-adiponectin ratio in the pericoronary fat [8]. This therapeutic anti-inflammatory effect reduces MACE in PDM patients with acute myocardial infarction (AMI) [8].

Thus, we might speculate that the pericoronary fat in PDM might produce higher SGLT2 protein and leptin levels, with a reduction of SIRT6 levels and over-inflammation. Consequently, all these effects could lead to the development and destabilization of atherosclerotic plaques in coronary arteries with consequent worse clinical prognosis. Therefore, in the current study, we aimed to investigate the expression of SGLT2, SIRT6, leptin, and inflammatory markers expression in the pericoronary fat and the clinical prognosis at one year of follow-up comparing PDM to normoglycemic (NG) patients subjected to coronary artery bypass grafting (CABG) for acute myocardial infarction (AMI). Intriguingly, because the metformin therapy could regulate pericoronary fat inflammation and reduce (MACE) in PDM patients with AMI [8], here, we also evaluated the effects of metformin therapy on the pericoronary SGLT2, SIRT6, leptin levels, inflammatory markers expression, and 12-month prognosis in AMI-PDM patients, divided in metformin user vs. never-metformin users.

## 2. Materials and Methods

In a multicenter study, we screened a population of patients with uncomplicated AMI, according to criteria recommended by the American College of Cardiology [7]. To date, the AMI was confirmed by the diagnosis of an acute myocardial injury with clinical evidence of acute myocardial ischemia and rise and/or fall of cardiac troponin (cTn) values with at least one value above the 99th percentile upper reference limit (URL) [7]. We obtained the routine analyses for these patients before performing coronary angiography, and we started the full medical therapy, including beta-blocker drugs and/or calcium antagonists, low-dosage aspirin, nitrates, and heparin before CABG [7]. Thus, the patients with AMI received an acute angiography of coronary vessels [7,8,9]. After that, the patients diagnosed with multi-vessel coronary artery disease were referred for CABG, according to international guidelines [7]. After CABG, the AMI patients entered prospectively into a database. The authors categorized the participants as PDM vs. patients with normal glucose (NG). PDM was defined by impaired fasting glucose, impaired glucose tolerance, and glycated hemoglobin A1c (HbA1c) values ≥5.7% but <6.5% [8,9,10]. The patients with PDM answered a specific questionnaire about the use of metformin before the beginning of the study, the beginning and the end of treatment, the administration route, and the duration of use [10]. Among PDM, the patients who never used metformin were classified as “non-metformin-users”. The PDM who had already used metformin for at least six months were classified as “metformin-users.”

The study population was recruited from the Department of Cardiovascular Surgery of the University of Campania “Luigi Vanvitelli, Naples, Italy, from the Department of Cardiovascular Surgery of the Santissima Annunziata Hospital, Sassari, Italy, and from the Department of Advanced Medical and Surgical Sciences, the University of Campania “Luigi Vanvitelli,” Naples, Italy, from January 2016 to June 2018. The study was approved by the local Ethics Committee, and informed written consent was obtained for each patient. The study was performed in accordance with the Declaration of Helsinki. The study endpoints were evaluated in the study cohorts after CABG at one year of follow-up. The enrolled patients respected the following inclusion and exclusion criteria.

### 2.1. Study Inclusion and Exclusion Criteria

The inclusion criteria were: patients aged >18 and <75 years with indication to receive a CABG for multi-vessel coronary artery stenosis; patients without a confirmed diagnosis of diabetes mellitus; patients without chronic inflammatory disease; patients without the neoplastic disease.

Exclusion criteria were: diagnosis of diabetes mellitus; patients with clinical or laboratory evidence of heart failure with New York Heart Association (NYHA) Class III or IV; patients with the previous CABG; patients with the previous stroke; valvular heart defects; malignant neoplasms; severe uncontrolled hypertension (blood pressure >200/100 mmHg) or secondary causes of hypertension; routinely consuming more than three alcoholic drinks per day; uncontrolled endocrine or metabolic disease known to influence glycemia (i.e., secondary causes of hyperglycemia); kidney failure with estimated glomerular filtration rate (eGFR) <30 mL/min/1.73 m2.2.2. 

### 2.2. Laboratory Analysis

The authors, blinded to the study protocol, measured, after an overnight fast, the plasma glucose, HbA1c, and serum lipid levels by enzymatic assays in the hospital chemistry laboratory. To date, we collected venous blood samples for troponin I (Behring Diagnostics, Westwood, MA, USA) in ethylene diamine tetra acetic acid-coated tubes immediately after patients arrived at the emergency department. Troponin T was measured with an Opus Magnum device (Behring Diagnostics, Frankfurt, Germany), with a discriminator value recommended by the manufacturer. Then, we determined the levels of fasting blood glucose before CABG. Fasting and postprandial plasma glucose data were obtained from the average of each assessment. 

### 2.3. Pericoronary Fat Excision: SLGT2, SIRT6, and Leptin Assay

During the CABG procedure, two experienced physicians blinded to study protocol and the study population’s characteristics removed the pericoronary fat [8]. They selected the epicardial coronary portion for the post-stenotic anastomosis [8]. This portion of coronary vessel was selected and isolated by removing pericoronary fat, as previously reported [8]. Thus, the pericoronary fat tissue (100 mg) was cut into small pieces and homogenized by using 300 µL of 2D lysis buffer (7 mM urea, 2 mM thiourea, 4% CHAPS [3-([3-cholamidopropyl/dimethylammonium)-1-propane sulfonate] buffer, 30 mM Tris-HCl, pH 8.8). SGLT2, SIRT6, and leptin levels in 100 μg of the protein extract from pericoronary fat specimens were assessed using specific human enzyme-linked immunosorbent assay (ELISA), MBS164261, MBS162109, and MBS020274, respectively (all from MyBioSource.com, accessed on 23 June 2021), according to manufacturers’ protocols. Briefly, duplicate standard points by serially diluting each standard stock solution were prepared. A total of homogenate about 40 µL, together with 10 µL anti-SGLT2, -SIRT6, or -leptin antibody, were added in each well while 50 µL/well of dilution were adjoined for standard curves. Streptavidin–horseradish peroxidase (HRP) conjugate (50 µL) was added to all wells and plate incubated for 1 h at 37 °C. After multiple washes, chromogen substrate solutions were added to each well and plate incubated once again for 10 min at 37 °C in the dark. Finally, colorimetric reaction was reached by the addition of stop solution (50 µL) and the optical density (O.D.) values recorded using a microplate reader (model 680 Bio-Rad, Milan, Italy) set to 450 nm. SGLT2, SIRT6, and leptin levels (ng/mL) in tissue homogenates were assessed by plotting sample O.D. values against the proper standard curve, as previously reported [8]. Each determination was repeated at least three times. The expression level of SLGT2 and SIRT6 was normalized with glyceraldehyde 3-phosphate dehydrogenase (GAPDH) as internal control using Image J software (1.52n software, National Institutes of Health) and values expressed as arbitrary units (AU). The loading control GAPDH used in gel electrophoresis and Western blotting studies, was a “housekeeping” protein, abundantly distributed in cells, and used to check the even loading of gel samples, and the even transfer of proteins at the blotting stage.

### 2.4. Immunoblotting Analysis

Pericoronary fat homogenate tissues from NG, PDM, PDM never-metformin-users, and PDM current-metformin-users patients were obtained as above reported, and immunoblot analyses determined the SIRT6 and SGLT2. Samples were lysed in a buffer containing 1% Nonidet *p*-40 for 30 min at 4 °C. The lysates were then centrifuged for 10 min at 10,000× *g* at 4 °C. After centrifugation, 50 μg of each sample were loaded, electrophoresed in polyacrylamide gel, and electroblotted onto a nitrocellulose membrane by Trans-Blot Turbo Transfer System (Bio-Rad, Milan, Italy). Membranes were incubated overnight at 4 °C with antibodies against SIRT6 and SGLT2 (ab62739 and ab37296, respectively, both 1:1000, from Abcam, Cambridge, UK) followed by incubation with secondary antibody. Marker signals were normalized with antibodies against anti-β-actin (ab8227, 1:5000, Abcam) or -α-tubulin protein (ab15246, 1:5000, Abcam). Immunoreactive signals were detected with a horseradish peroxidase–conjugated secondary antibody, detected by Excellent Chemiluminescent Substrate kit (E-IR-R301, Elabscience) and visualized by using ChemiDoc Imaging System with Image Lab 6.0.1 software (Bio-Rad Laboratories, Milan, Italy). After background subtraction, the densities of immunoreactive bands were measured with Image J software (National Institutes of Health), and expressed as arbitrary units (AU). Each western blot was repeated at least three times.

### 2.5. Detection of Inflammatory Cytokine and 3-Nitrotyrosine Levels

The assessment of some pivotal cytokine levels involved in the inflammatory status (interleukin-6 (IL-6), interleukin-1beta (IL-1β), tumor necrosis factor-alpha (TNF-α), and monocyte chemoattractant protein 1 (MCP-1)) and 3-nitrotyrosine levels as a marker of protein damage by oxidative stress was performed with specific ELISA assays (Cymax^TM^ Human IL-6, AbFrontier, LF-EK0260; Cymax^TM^ Human IL-1β, AbFrontier, LF-EK0276; CymaxTM Human TNFα, AbFrontier, LF-EK0193; human MCP-1 ELISA, BioVendor, RAF081R; Abcma, ab116691, respectively) according to the manufacturer’s protocols. Adequate volumes (approximately 100 µL) of pericoronary fat homogenates from NG, PDM, PDM never-metformin-users, and PDM current-metformin-users patients were incubated in microplate wells pre-coated with specific anti-cytokine or anti-3-nitrotyrosine detector antibody. After 1 h incubation and washing to remove unbound cytokines, biotin-labeled anti- IL-6, -IL-1β, -TNF-α, and -MCP-1 antibodies were added and incubated for an additional hour. After another washing, the streptavidin-HRP conjugate is added and incubated for 30 min. The absorbance for all analyzed parameters is measured at 450 nm using a microplate reader model 680 Bio-Rad and cytokine or 3-nitrotyrosine levels estimated by plotting sample absorbance values against respective standard curves.

### 2.6. Clinical Visits, Data Collection, and Analysis

At baseline and follow-up, the involved physicians evaluated the study population’s clinical characteristics as NG vs. PDM and PDM-metformin users vs. PDM-non metformin users. Furthermore, we collected and analyzed the data during clinical visits 10 days after clinical discharge. At 1 year of follow-up by the treating physician, telephonic interviews, hospital admissions, and discharge schedules [5,6,7]. Moreover, at the follow-up visits, the physicians blinded to study protocol evaluated each patient’s clinical status and performed a physical examination with the collection of vital signs and adverse events [5,6,7]. Again, the physicians evaluated the adherence to drug therapy and any clinical symptom referred by any patient [5,6,7]. Finally, the authors evaluated the MACE at follow-up end, collecting the data prospectively from electronic medical records used in the clinical setting at participants’ Institutions. An electronic system was used to capture, collect, and monitor the data, with on-site and real-time data entry. Finally, the authors collected the patients’ files in each participating Institution that were then analyzed. 

### 2.7. Major Adverse Cardiac Events Definition

The authors defined the MACE as a composite endpoint indicating cardiovascular disease events, hospital admissions for heart failure, and ischemic cardiovascular events. We diagnosed the cardiovascular disease events by evidence of ischemic heart disease, peripheral arterial disease, stroke/transitory ischemic attack, or revascularization procedure [7,10]. Finally, the study investigators reported all the events with the potential to be adjudicated as one of the predefined study endpoints, regardless of the investigator’s opinion [10]. In identifying a suspected unreported event by a reviewer, the authors asked the reviewer to make a note back to the investigator [11]. Thereafter, the authors collected the MACE during patients’ interviews, visits, and hospital discharge schedules [11].

### 2.8. Study Endpoints

The study’s primary endpoint was the evaluation of MACE at 12-months of follow-up in PDM vs. NG, and in PDM metformin users vs. PDM never-metformin users with AMI.

The study’s secondary endpoint was the evaluation of SGLT2, leptin, SIRT6 expression, and inflammatory markers in pericoronary fat (before CABG) of PDM vs. NG PDM metformin users vs. PDM never-metformin users with AMI.

All patients underwent quarterly clinical evaluation and routine analyses as outpatients for 12 months after the event. The ethics committee approved the clinical protocol (number 371).

### 2.9. Statistical Analysis

SPSS version 23.0 (IBM statistics) was used for all statistical analyses. Categorical variables were presented as number and percentage, whilst continuous variables as either mean ± standard deviation or median and interquartile range, in the case of not normally distributed variables. The normal/not normal distribution was preliminarily assessed through the Kolmogorov–Smirnov goodness of fit K–S test.

For comparison among study populations, propensity score matching (PSM) was developed from the predicted probabilities of a multivariable logistic regression model predicting mortality, and events from age, sex, body mass index (BMI), waist–hip ratio (WHR), ST-elevation, and No-ST-elevation myocardial infarction, current smoking, family history, hypertension, dyslipidemia, smoking history, high density lipoprotein (HDL)-cholesterol, low density lipoprotein (LDL)-cholesterol, triglycerides levels, beta-blockers, ace inhibitors, calcium inhibitors, thiazide diuretics, aspirin, vessel disease, and coronary lesions. 

The MACE (death, re-acute coronary syndrome) rates were derived as Kaplan–Meier estimates and compared by log-rank test at 1 year of follow-up. Overall survival and event-free survival were assessed by Kaplan–Meier survival curves and compared by the log-rank test. The resulting hazard ratios (HRs), and 95% CIs were reported. Two-tailed *p* values < 0.05 were considered statistically significant. 

## 3. Results

Three hundred and sixty NG patients and 240 PDM met inclusion criteria. After PSM for anthropometric characteristics and cardiovascular risk factors, a total of 150 NG-patients were matched to 150 PDM (Table 1). Among PDM, 66 were current-metformin-users, and 174 were never-metformin-users. During 1-year follow-up, 13 PDM (10 never-metformin-users and three current-metformin-users) developed type 2 diabetes and were excluded from the study. After PSM, 55 current-metformin-users (670 ± 270 mg/daily) were matched to 55 never-metformin-users. The mean (±SD) duration of metformin treatment was 20 ± 6.3 months. 

From the tissue analysis of pericoronary fat, both immunoblotting analysis and ELISA assay revealed that SGLT2 content and protein expression levels were significantly higher in PDM pericoronary fat specimens than NG-patients (*p* < 0.01), and lower in PDM under metformin therapy vs. never-metformin users (*p* < 0.01) (Figure 1). 

Notably, compared to the NG-patients, the PDM showed a decreased SIRT6 tissue content and protein expression levels, which was lowest in tissues from PDM never-metformin-users vs. PDM metformin-users (*p* < 0.01) (Figure 2).

As for the SGLT2 content, leptin levels were significantly higher in pericoronary fat specimens from PDM vs. NG patients (*p* < 0.01) (Figure 3). Of interest, with respect to the never-metformin-users group, the metformin-users group presented a significantly lower leptin expression (*p* < 0.01).

Intriguingly, we found higher 3-nitrotyrosine and inflammatory cytokine levels in PDM vs. NG-patients (*p* < 0.01) (Figure 3). Among PDM patients, 3-nitrotyrosine and TNF-α, MCP-1, IL-6, and IL-1β levels were significantly higher in never-metformin-users than in metformin users (*p* < 0.01) (Figure 3).

Furthermore, at linear regression analysis, we found that the values of pericoronary fat leptin contents (dependent variable) changed when SGLT2 levels (independent variables) were varied (R^2^ = 0.7765, *p* < 0.05). In contrast, the other independent variables were held fixed (Figure 4).

Finally, the MACE was 12% in all PDM vs. 4% in NG-patients (*p* < 0.01); the PDM metformin-users vs. never-metformin users presented a significantly lower rate of MACE (6.9% vs. 20.7%; *p* < 0.01) at the follow-up end. The Kaplan curve shows the different survival free from MACE in NG patients vs. PDM patients (*p* < 0.05), and in PDM patients with metformin vs. PDM patients without metformin therapy (*p* < 0.05) at 1 year of follow-up (Figure 5). Finally, to translate the role of pericoronary fat inflammatory tone in real clinical endpoints, we evaluated MACE stratified by SGLT2 and leptin levels. As evidenced in Figure 5, patients with higher SGLT2 and leptin levels had a higher number of events (*p* < 0.05).

## 4. Discussion

In the current study, we showed that the PDM vs. NG-patients over-expressed the SGLT2 and leptin at the level of pericoronary fat. Notably, SGLT2 and leptin expression in the pericoronary fat was higher in the PDM never-metformin-users vs. PDM metformin-users. Intriguingly, the expression of leptin and SGLT2 in the pericoronary fat were strictly linked, and the leptin contents changed at the variation of SGLT2 levels. This negative trend in PDM vs. NG patients, particularly in PDM metformin users vs. PDM never-metformin users, was linked to significantly lower expression of SIRT6 and over-inflammation at the level of the pericoronary fat.

At the clinical level, PDM vs. NG, and PDM never-metformin-users vs. PDM metformin-users evidenced a higher rate of MACE at 1 year of follow-up.

Moreover, summarizing the study results, we might say that the (PDM) patients with higher SGLT2 and leptin levels, as those with lowest SIRT6 and over-inflammation, had a higher rate of MACE. Secondly, the MACE was significantly reduced by metformin therapy in PDM patients.

To the best of our knowledge, here, for the first time, we suggested that the pericoronary fat levels of leptin and SGLT2 could be amplified during AMI. Notably, their over-expression might be responsible for a worse prognosis in PDM vs. NG patients, particularly those without metformin therapy. Indeed, in the current study, we translated at the clinical level (MACE rate) the complex metabolic/molecular alterations occurring during AMI in PDM vs. NG patients. Intriguingly, we investigated these abnormalities by the pericoronary fat expression of SGLT2 and leptin levels and by the known role played by SIRT6/over-inflammation [5,8,12,13]. Notably, SGLT2 and leptin were mainly expressed in the pericoronary fat of PDM as compared to NG. In addition, SLGT2/leptin were over-expressed in PDM with poor glycemic control and insulin resistance (never-metformin users vs. metformin-users). In this context, we know that, in AMI patients, the pericoronary fat expression of leptin could be modulated by SGLT2 activity [3].

Conversely, we showed that the patients with higher pericoronary fat SGLT2 levels had a higher number of MACE. Furthermore, we reported a protective effect played by metformin therapy in AMI patients with PDM. Indeed, metformin caused the amelioration of cardiovascular outcomes in PDM after AMI (reduction of MACE). Intriguingly, the metformin exerted its effects via the significant reduction of both leptin and SGLT2 expression at the level of the pericoronary fat. This point appears to be crucial in the current investigation, and we might currently report on a more complex SLGT2/leptin axis in PDM patients with AMI. Indeed, we remark that, at first, SLGT2/leptin axis is implied in the MACE in post-AMI patients. Secondly, metformin therapy might induce the amelioration of clinical outcomes post-AMI in PDM patients via the regulation of SLGT2/leptin axis, via modulation of SIRT6, and downregulation of the inflammatory axis.

Notably, metformin is a hypoglycemic drug that reduces glycemia, insulin resistance, and overweight in PDM patients [12,13]. Indeed, to reduce glycemia and insulin resistance, metformin plays a role as a pleiotropic effector of adipocytes’ inflammatory and metabolic functions [12,13]. Intriguingly, all of these metabolic and anti-inflammatory effects ameliorated cardiac performance and clinical outcomes in PDM patients [11,12,13,14,15]. In addition, metformin is a modulator of the coronary endothelial function [15]. Indeed, metformin therapy resulted in the amelioration of coronary endothelial function in PDM patients [15]. Thus, metformin could regulate the metabolic effectors linked to coronary endothelial dysfunction, restenosis, and MACE after coronary revascularization in PDM patients [15].

On the other hand, here we evidenced that metformin therapy reduced SGLT2 expression levels in the pericoronary fat of AMI patients with PDM. Consequently, this effect might reduce the MACE in metformin vs. never-metformin users. This result represents a novelty, and it has never been investigated before in humans. Indeed, from the current literature, we know that the inhibition of SGLT2 resulted in the reduction of mortality rate after AMI in a rat model [16]. Indeed, the inhibition of SGLT2 in rats by empagliflozin, an SGLT2 inhibitor, prevented the reduction of ATP level in the non-infarcted myocardium after AMI via the increase of sirtuins’ myocardial levels and superoxide dismutase 2 [16]. Intriguingly, the empagliflozin prevented the diabetes-induced increase in post-AMI mortality, acting by protective modification of cardiac energy metabolism and antioxidant proteins [17]. At the same level, another SGLT2 inhibitor, the canagliflozin, caused either a glucose-independent upregulation of cardiac survival pathways [17]. Thus, this resulted in a cardioprotective intervention in high-risk cardiovascular patients irrespective of diabetic status [17].

Therefore, we might speculate that metformin’s anti-inflammatory and metabolic properties could affect the SGLT2 expression at the level of pericoronary fat. To date, we might speculate that metformin might indirectly inhibit the SGLT2 expression at the level of pericoronary fat in PDM patients with AMI. Again, metformin could also modulate the leptin expression (at the level of pericoronary fat). In this way, metformin might be marked as the regulative drug of SLGT2/leptin axis in PDM patients with AMI. In our opinion, these results might have important clinical implications. Furthermore, we might recommend metformin therapy in PDM patients with AMI. Indeed, the metformin therapy could have the double effect of reducing the progression to type 2 diabetes [10], and the MACE in PDM patients with AMI. Thus, we might recommend using metformin therapy in high-risk patients as those with PDM. Indeed, metformin is prescribed for only 3.7% of PDM in the current clinical practice despite ADA recommendation [10].

The present study evidenced few limitations. The small sample size and short duration of follow-up could reduce the impact of the reported data. Conversely, the loss of an ex vivo in vitro model cannot drive us towards a definitive explanation of the pericoronary fat expression of leptin/SGLT2 in different conditions of glycemic overload treatment metformin. Further studies in a larger population, at a more extended time follow-up duration, are needed, to investigate all these molecular, cellular, and clinical effects in PDM patients with AMI.

## 5. Conclusions

The current study evidenced the existing link between the pericoronary fat expression of SGLT2, leptin, SIRT6, over-inflammation, and worse clinical outcomes. Notably, and clinically relevant, these pathways were mainly expressed in PDM vs. NG and modulated by the metformin therapy in PDM. Indeed, metformin therapy could reduce the pericoronary fat levels of the SGLT2/leptin axis. Notably, metformin therapy reduced the expression of SIRT6 and inflammatory cytokines at the level of pericoronary fat. Finally, it led to the amelioration of clinical outcomes.

Therefore, we might propose metformin therapy as a regulator of the pericoronary fat expression of SGLT2/leptin axis and to ameliorate PDM patients’ clinical outcomes.

## Figures and Tables

**Figure 1 biomedicines-09-00904-f001:**
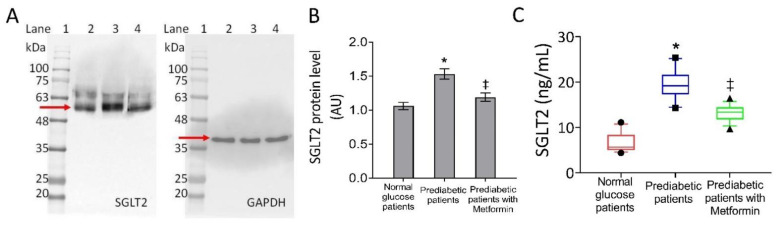
Sodium-glucose transporter 2 (SGLT2) levels in pericoronary fat patients. (**A**,**B**) SGLT2 protein level in pericoronary fat specimens from 150 normal glucose, 150 pre-diabetic patients, and 55 pre-diabetic current-metformin-users detected by western blotting analysis. Lane 1 = molecular markers; lane 2 = normal glucose patients; lane 3 = pre-diabetic patients; lane 4 = pre-diabetic patients with metformin. The expression level was normalized with glyceraldehyde 3-phosphate dehydrogenase (GAPDH) as internal control using Image J software and values expressed as arbitrary units (AU). (**C**) SGLT2 level in pericoronary fat specimens from 150 normal glucose, 150 pre-diabetic patients, and 55 pre-diabetic current-metformin-users assessed by enzyme-linked immunosorbent assay (ELISA) assay on homogenates. Boxplots represent the median, 25th, and 75th percentile values, while the black symbols outside represent the 10th and 90th percentiles. For comparison between the cohorts, we used ANOVA test. * *p* < 0.05 vs. normal glucose patients; ‡ *p* < 0.05 vs. pre-diabetic patients.

**Figure 2 biomedicines-09-00904-f002:**
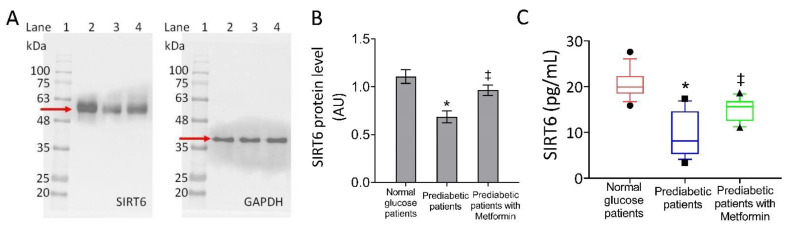
Sirtuin 6 (SIRT6) levels in pericoronary fat patients. (**A**,**B**) Sirt6 protein level in pericoronary fat specimens from 150 normal glucose, 150 pre-diabetic patients, and 55 pre-diabetic current-metformin-users detected by western blotting analysis. Lane 1 = molecular markers; Lane 2 = normal glucose patients; lane 3 = pre-diabetic patients; lane 4 = pre-diabetic patients with metformin. The expression level was normalized with glyceraldehyde 3-phosphate dehydrogenase (GAPDH) as internal control using Image J software and values expressed as arbitrary units (AU). (**C**) SIRT6 level in pericoronary fat specimens from 150 normal glucose, 150 pre-diabetic patients and 55 pre-diabetic current-metformin-users assessed by ELISA assay on homogenates. Boxplots represent the median, 25th and 75th percentiles values while the black symbols outside the 10th and 90th percentiles. For comparison between the cohorts we used the ANOVA test. * *p* < 0.05 vs. normal glucose patients; ‡ *p* < 0.05 vs. pre-diabetic patients.

**Figure 3 biomedicines-09-00904-f003:**
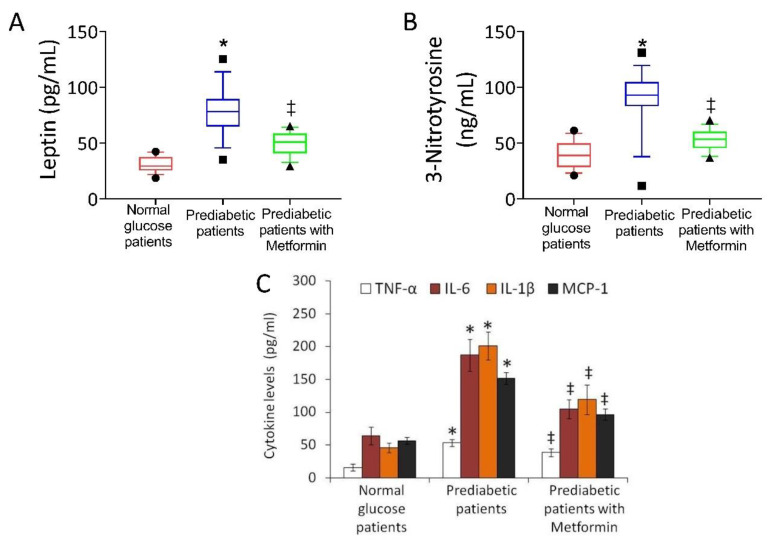
Leptin levels and inflammatory status in pericoronary fat patients. (**A**) Leptin and (**B**) 3-nitrotyrosine level in pericoronary fat specimens from 150 normal glucose, 150 prediabetic patients, and 55 prediabetic current-metformin-users assessed by enzyme-linked immunosorbent assay (ELISA) assay on homogenates. Boxplots represent the median, 25th, and 75th percentiles values, while the black symbols outside represent the 10th and 90th percentiles. (**C**) Assessment of cytokine levels in pericoronary fat specimens from 150 normal glucose, 150 prediabetic patients, and 55 prediabetic current-metformin-users assessed by the ELISA assay on homogenates. TNFα: tumor necrosis alpha; IL-6: interleukin 6; IL-1β: interleukin 1 beta; MCP-1: monocyte chemoattractant protein 1. For comparison between the cohorts, we used the ANOVA test. * *p* < 0.05 vs. normal glucose patients; ‡ *p* < 0.05 vs. prediabetic patients without metformin.

**Figure 4 biomedicines-09-00904-f004:**
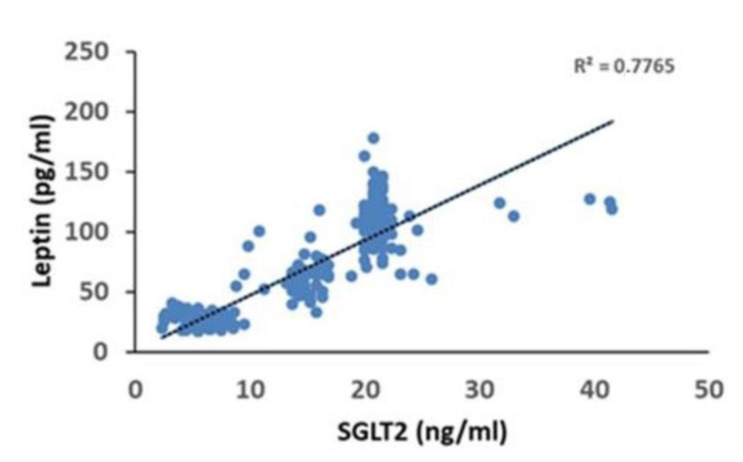
Regression analysis evidences a relationship between pericoronary fat leptin contents and sodium-glucose transporter 2 (SGLT2) in the overall study population.

**Figure 5 biomedicines-09-00904-f005:**
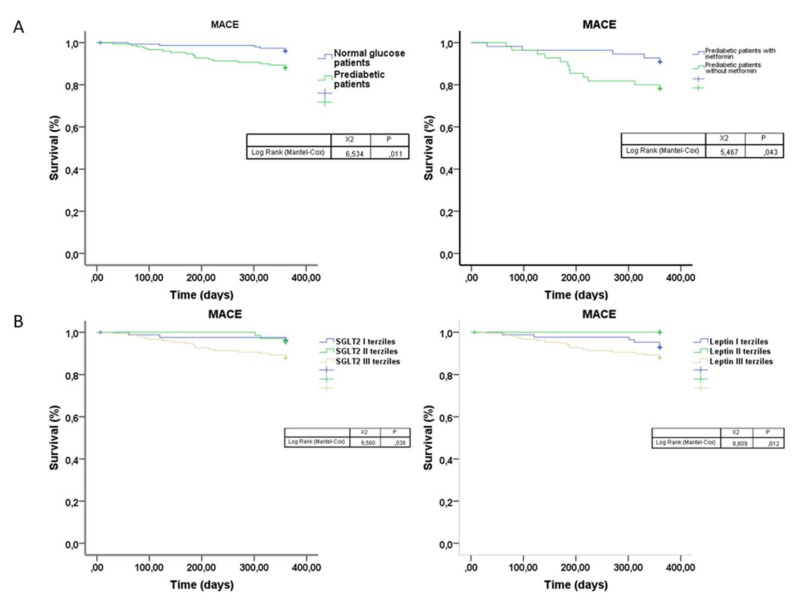
(**A**) In the left (upper part), the Kaplan–Meier survival curves free from major adverse cardiac events (MACE) in normal glucose patients (blue color) vs. pre-diabetic patients (green color) at 1 year of follow-up. In the right (upper) part, the Kaplan–Meier survival curves free from MACE in pre-diabetic patients with metformin (blue color) vs. pre-diabetic patients without metformin therapy (green color) at 1 year of follow-up. (**B**) In the left (inferior) part, the Kaplan–Meier survival curves free from MACE, according to SGLT2 tertiles at 1 year of follow-up. In the right (inferior) part, the Kaplan–Meier survival curves free from MACE, according to leptin tertiles at 1 year of follow-up.

**Table 1 biomedicines-09-00904-t001:** Baseline clinical characteristics of patients with acute myocardial infarction (AMI) matched by propensity score analysis.

	Normoglycemic Patients	Pre-Diabetic Patients	*p*	Prediabetic Never Metformin Users	Pre-Diabetic Current Metformin Users	*p*
Number	150	150		55	55	
Mean age (years)	68.2 ± 6.8	67.1 ± 6.3	0.082	66.2 ± 5.4	67.1 ± 4.9	0.345
Sex (M/F)	99/81	96/84	/	33/25	34/24	/
BMI (kg/m^2^)	27.1 ± 1.6	27.4 ± 1.9	0.088	27.3 ± 0.7	26.9 ± 1.2	0.140
Systolic blood pressure (mmHg)	130.8 ± 13.1	132.9 ± 10.9	0.064	131.2 ± 8.4	133.1 ± 7.7	0.945
Diastolic blood pressure (mmHg)	79.2 ± 6.5	79.3 ± 6.6	0.898	77.7 ± 6.4	79.4 ± 6.7	0.154
Heart rate (bpm)	87.2 ± 8.1	86.9 ± 8.6	0.744	85.9 ± 5.6	87.2 ± 8.8	0.357
**Prediabetes diagnosis criteria**						
Fasting plasma glucose, *n* (%)	/	39 (21.6)	/	10 (17.2)	11 (18.9)	0.488
Post-prandial glucose, *n* (%)	/	18 (10)	/	11 (18.9)	10 (17.2)	0.502
HbA1c, *n* (%)	/	102 (56.7)	/	24 (41.4)	25 (43.1)	0.493
Two or more criteria, *n* (%)	/	21 (11.6)	/	13 (22.4)	12 (20.7)	0.559
**Risk Factors**						
Hypertension, *n* (%)	143 (79.4)	144 (80.1)	0.500	49 (84.5)	51 (87.9)	0.394
Hyperlipemia, *n* (%)	112 (62.2)	110 (61.1)	0.457	49 (84.5)	47 (81.1)	0.403
Cigarette smoking, *n* (%)	141 (78.3)	142 (78.9)	0.500	47 (81.1)	51 (87.9)	0.221
**Active treatments**						
beta-blockers, *n* (%)	104 (57.8)	106 (58.9)	0.457	43 (74.1)	41 (70.7)	0.418
ACE inhibitors, *n* (%)	76 (42.2)	74 (41.1)	0.457	22 (37.9)	27 (46.6)	0.226
Angiotensin receptor blockers, *n* (%)	88 (48.9)	75 (41.7)	0.102	25 (43.1)	22 (37.9)	0.353
Calcium inhibitor, *n* (%)	100 (55.6)	105 (58.3)	0.335	37 (63.8)	41 (70.7)	0.277
Nitrate, *n* (%)	46 (25.6)	34 (18.9)	0.081	7 (12.1)	9 (15.5)	0.394
Statins, *n* (%)	92 (51.1)	100 (55.6)	0.230	32 (55.2)	28 (48.3)	0.289
Thiazide diuretic, *n* (%)	55 (30.6)	56 (31.1)	0.500	22 (37.9)	25 (43.9)	0.353
Aspirin, *n* (%)	115 (63.9)	118 (65.6)	0.413	37 (63.8)	38 (65.5)	0.500
Thienopyridine, *n* (%)	24 (13.3)	24 (13.3)	0.562	6 (10.3)	7 (12.1)	0.500
**Laboratory analyses**						
Plasma glucose (mg/dL)	86.5 ± 6.5	110 ± 7.3	0.001	111.5 ± 9.3	110.9 ± 7.6	0.762
HbA1c (%)	5.1 ± 0.3	5.9 ± 03	0.001	6.2 ± 0.3	6.1 ± 0.4	0.463
Cholesterol (mg/dL)	205.9 ±21.1	205.1 ± 22.1	0.670	203.2 ± 19.1	206.6 ± 19.5	0.353
LDL-cholesterol (mg/dL)	127.4 ± 20.9	131.3 ± 21.5	0.080	128.6 ± 29.4	133.1 ± 18.9	0.198
HDL-cholesterol (mg/dL)	38.3 ± 3.5	38.1 ± 3.5	0.764	37.8 ± 3.3	37.5 ± 3.5	0.587
Triglycerides (mg/dL)	181.5 ± 19.6	182.6 ± 22.1	0.621	185.1 ± 23.6	179.6 ± 17.3	0.154
Creatinine (mg/dL)	0.99 ± 0.13	0.98 ± 0.19	0.399	0.98 ± 0.17	0.97 ± 0.16	0.833
hs-cTnT (ng/L)	147.6±32.7	149.6 ± 25.9	0.411	149.9 ± 32.1	149.2 ± 26.6	0.604

BMI: body mass index; Hb1Ac: type 1Ac glycated hemoglobin; ACE: angiotensin converting enzyme; LDL: low density lipoprotein; HDL: high density lipoprotein; hs-cTnT: elevated high sensitivity cardiac troponin T. Data are means ± SD or *n* (%); *p* < 0.05 is statistical significant.

## Data Availability

The datasets analyzed during the current study are not publicly available because of unpublished data, but are available from the corresponding author upon reasonable request.

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
