# Peer review of "Metformin Therapy Effects on the Expression of Sodium-Glucose Cotransporter 2, Leptin, and SIRT6 Levels in Pericoronary Fat Excised from Pre-Diabetic Patients with Acute Myocardial Infarction"

_biomedicines, 2021, doi:10.3390/biomedicines9080904_

Round 1

Reviewer 1 Report

There are few misuesed words, mistyping and punctuation errors in the text.

Explane abbreviations when they are met first time, e.g. PSM.

What is WHI? Is it WHR? (waist to hip ratio)

How about taking anti inflammatory drugs (e.g. NSAI agents) by the study subjects? Could  NSAI agents’ use modulate the obtained results?

In the References – the literature source repeats itself 2 times (sources 12. and 13.)

Sentence in Discussion (rows 397 – 400) could to be formulated more clearly.

Author Response

#REVIEWER 1

Comments and Suggestions for Authors

There are few misuesed words, mistyping and punctuation errors in the text.

We thank the reviewer for the helpful comment. As you suggested we revised the misused words, mistyping and punctuation errors in the text.

Explane abbreviations when they are met first time, e.g. PSM.

As suggested, we explained the abbreviations in the text.

What is WHI? Is it WHR? (waist to hip ratio).

We are sorry for the mistake. It is WHR. Thus, we replaced WHI with the WHR.

How about taking anti inflammatory drugs (e.g. NSAI agents) by the study subjects? Could  NSAI agents’ use modulate the obtained results?

We really appreciate this relevant observation. Indeed, the anti-inflammatory drugs as NSAI agents could reduce the inflammatory burden, added to anti-platelets effects. However, numerous studies have been conducted to well address this point. In this setting, recently authors conducted a meta-analysis including a cohort of 446,763 patients taking any dose of NSAI for 1 week, 1 month, or >1 month (BMJ 2017; 357. doi: 10.1136/bmj.j1909). Intriguingly, they showed for these patients an increased risk of MI (BMJ 2017; 357. doi: 10.1136/bmj.j1909). Notably, the probability of increased MI risk was 92% for celecoxib, 97% for ibuprofen, and 99% for diclofenac, naproxen, and rofecoxib, with corresponding odds ratios of 1.24 (0.91-1.82) for celecoxib, 1.48 (1.00-2.26) for ibuprofen, 1.50 (1.06-2.04) for diclofenac, 1.53 (1.07-2.33) for naproxen, and 1.58 (1.07-2.17) for rofecoxib (BMJ 2017; 357. doi: 10.1136/bmj.j1909). Conversely, a greater risk of MI was documented for higher dose of NSAIDs (BMJ 2017; 357. doi: 10.1136/bmj.j1909). Moreover, these authors conclude that all NSAIDs, increased the risk of acute myocardial infarction (BMJ 2017; 357. doi: 10.1136/bmj.j1909).

Taking in observation these results, we would to remark the following concepts observed in our study:

-firstly, in our study 11 (7.3%) of NG vs. 14 (9.3%) of PDM were under NSAI (p=0.677). Notably, after PSM the metformin-users vs. non-users showed a (non-significant) higher percentage of NSAI agents taking (7 (12.7%) vs. 5 (9.1%); p=0.223).

-Despite a (non-significant) higher use of NSAI agents in PDM vs. NG, and in PSM metformin-users vs. non-users, we found that PDM vs. NG expressed higher peri-coronary levels of 3-nitrotyrosine and inflammatory cytokine (TNF-α, MCP-1, IL-6, and IL-1β levels; p<0.05). Again, as reported the inflammatory markers were significantly higher in never-metformin-users than in metformin users (p<0.01), (Figure 3).

-Finally, despite a (non-significant) higher use of NSAI agents in PDM vs. NG, and in PSM metformin-users vs. never-users, we found that the MACE was 12% in all PDM vs. 4% in NG-patients (p<0.01), and 6.9% vs. 20.7% (p<0.01) comparing the PDM metformin-users vs. never-metformin users respectively.

However, we might conclude that, despite the over-inflammation in PDM never-metformin-users vs. other cohorts of study, and the evidence of not statistical different percentage of NSAI agents taking in NG, PDM and metformin-users vs. non-users, we found a significantly lower rate of MACE in NG vs. PDM, and in metformin-users vs. non-users (p<0.05) at follow-up end.

Therefore, according to our data, and to the evidence of recent studies and meta-analysis, we might conclude that NSAI agents did not affect MACE post AMI in overall study population, and specifically comparing NG vs. PDM and PMD metformin-users vs. non-users.

In the References – the literature source repeats itself 2 times (sources 12. and 13.)

We are sorry for the mistake. As suggested, we removed the reference 13.

Sentence in Discussion (rows 397 – 400) could to be formulated more clearly.

We fully agree with your comment. As you suggested, we re-wrote the period in a more clear way.

(x) English language and style are fine/minor spell check required

As suggested, we revised the English language and style (fine/minor spell check) by a native English mother tongue.

Reviewer 2 Report

In this interesting and pertinent study, the authors demonstrate a promising effect of metformin therapy by reducing inflammatory parameters in pre-diabetic patients and consequent amelioration of the conditions of acute myocardial infarction. At the same time, they establish a previously unreported link between the ­expression of SGLT2, leptin, SIRT6 and over-inflammation in pre-diabetic patients. In specific, in this study the authors tested the occurrence of major-adverse-cardiac-arrests in patients with pre-diabetes compared to non-glycemics, and patients using metformin with patients who never used metformin.  For the patients with pre-diabetes, metformin users had a significantly lower occurrence of major adverse cardiac events during the study. The authors suggest this was related to higher levels of inflammatory cytokines, SGL2 and leptin content and lower levels of SIRT6 protein levels in patients with pre-diabetes compared with normoglycemic patients. In order to attest this previously unreported mechanisms and relations, the authors employed a coherent and comprehensive laboratorial approach, by analyzing pericoronary fat tissues with subsequent immunoblotting analysis, detection of inflammatory cytokine levels. As a whole, the findings of this work will likely have important consequences for the improvement of the cardiovascular outcomes of patients with pre-diabetes and for the mechanisms of pericoronary fat over-inflammation.

As for the writing of manuscript, the objectives of the study (study endpoints) are stated very clearly by the authors and their pertinence is compared in a satisfactory way with past research of the respective scientific area. Overall, the quality of the writhing is quite high, and the use of English is appropriate. The methodological section is adequate, comprehensive and allows for replication studies, in particular, the employed statistical analysis is adequate, so are the chosen baseline clinical characteristics of patients. The limitations of the manuscript are quite minor, and are restricted to the occasional lack of some details in the methodological section, minor incorrections, and the need to upload higher resolution versions of some figures and eventual substitution of other figure.  As for the specific comments for the manuscript they are included in the attached word file.

Author Response

#REVIEWER 2

Comments and Suggestions for Authors

In this interesting and pertinent study, the authors demonstrate a promising effect of metformin therapy by reducing inflammatory parameters in pre-diabetic patients and consequent amelioration of the conditions of acute myocardial infarction. At the same time, they establish a previously unreported link between the ­expression of SGLT2, leptin, SIRT6 and over-inflammation in pre-diabetic patients. In specific, in this study the authors tested the occurrence of major-adverse-cardiac-arrests in patients with pre-diabetes compared to non-glycemics, and patients using metformin with patients who never used metformin.  For the patients with pre-diabetes, metformin users had a significantly lower occurrence of major adverse cardiac events during the study. The authors suggest this was related to higher levels of inflammatory cytokines, SGL2 and leptin content and lower levels of SIRT6 protein levels in patients with pre-diabetes compared with normoglycemic patients. In order to attest this previously unreported mechanisms and relations, the authors employed a coherent and comprehensive laboratorial approach, by analyzing pericoronary fat tissues with subsequent immunoblotting analysis, detection of inflammatory cytokine levels. As a whole, the findings of this work will likely have important consequences for the improvement of the cardiovascular outcomes of patients with pre-diabetes and for the mechanisms of pericoronary fat over-inflammation.

We thank the reviewer for the positive comment.

As for the writing of manuscript, the objectives of the study (study endpoints) are stated very clearly by the authors and their pertinence is compared in a satisfactory way with past research of the respective scientific area. Overall, the quality of the writhing is quite high, and the use of English is appropriate. The methodological section is adequate, comprehensive and allows for replication studies, in particular, the employed statistical analysis is adequate, so are the chosen baseline clinical characteristics of patients. The limitations of the manuscript are quite minor, and are restricted to the occasional lack of some details in the methodological section, minor incorrections, and the need to upload higher resolution versions of some figures and eventual substitution of other figure.  As for the specific comments for the manuscript they are included in the attached word file.

According to reviewer comment, we revised the suggested points in methodological section, and we update higher resolution version of all figures. Then, we replied to all included specific comments for the manuscript.

Comments

Line 57: “worse prognosis”: Which worse prognosis, specifically.

We specified this point, and replace “worse prognosis” with “higher rate of Major Adverse Cardiac Events (MACE)”.

Line 58: “fewer data have been reported about the expression of SGLT2 protein in the ectopic fat of PDM patients”. Any reference to support this statement?

We really appreciate your observation. On the other hand, in literature there are not data to support this sentence. Thus, we might suggest that actually our study might be seen as the first research conducted to evaluate the expression of SLGT2 protein in the ectopic fat of PDM patients.

Line 76: Although this is discussed later in the manuscript, it could be useful to introduce to the reader what metformin is.

We addressed this point in the Introduction (page 2, lines 70-73).

Line 85: Please specify what “CTn” is.

We are sorry for the mistake. We specified it in the text.

Line 135: How many total samples/individuals were subjected to this procedure?

For this procedure we excised multiple samples of the pericoronary fat from all NG and PDM patients enrolled in the study.

Line 151: Please add a reference to the microplate reader.

We added it.

Line 156: What was the total number of tissue samples?

We had 6 independent tissue lysates performed in triplicate for each enrolled patient.

Line 180: “adequate volumes” Which volumes approximately?

It was approximately 100 microliters of volume. We added it in the text.

Line 193: Why however, an adversative here. It would make more sense if it was “furthermore”.

We replaced however with furthermore.

Line 257: This section could be better introduced. There is no continuation from the previous sentence and results.

According to your observation we re-wrote this section.

Line 266: What is GAPDH in this context? It is not mentioned in the methods section.

The GAPDH is the Glyceraldehyde 3-phosphate dehydrogenase. We mentioned the GAPDH and its role in the coronary fat expression of SLGT2 and SIRT6 in the Methods, at page 4, lines 161-166.

Line 270: Which statistical test was used for this comparison? It should be mentioned here.

For comparison we used ANOVA test. We reported it in the figures’ legend.

Line 288: The figure is overlapping the legend text.

We corrected it.

Line 298: What sort of regression analysis was used? Linear regression analysis?

We used linear regression analysis.

Line 301/figure 4: The figure of a regression analysis does not seem to match the figure legend. There is no separation of green or blue colors. Neither any description to Leptin and SGLT2 concentrations.  It also does not match the description of the results above.

We are sorry for the mistake. We uploaded the correct version of figure 4 (regression curve) and figure 5 (Kaplan curves).

Line 312: “Figure 5”. The reference to this figure is decontextualized, it should be added properly to any of the previous or subsequent sentences.

We corrected this mistake.

Figure 5:  Please provide a figure with higher quality. It is not possible to read the figure legends nor to observe the graph lines. 

We uploaded a high quality figure 5.

Line 328: “at levelo of”. Please replace with “at level of”

We are sorry fot the mistake. We corrected it, replacing “at levelo of” with “at level of”.

Line 354: “we might currently report about a more complex SLGT2/leptin 354 axis in PDM patients with AMI”. Does this refer to future studies? Not clear.

7

Line 368: Why did the authors write “however” here?  These results seem to be in line with the previous results.

We removed however from the sentence.

Line 376: Empagliflozin should not be in capital letters, latter the authors write it without capital letters.

We are sorry for the mistake. We corrected it.

Line 397: “the loss of an ex vivo in vitro model”. Which model specifically? This was not clearly mentioned before. 

With this sentence we would say that in the present study we investigated only the expression of inflammatory markers, leptin and SLGT2 from humans’ tissues of peri-coronary fat. Thus, we did not have an ex vivo model, as cultured exerted human cells to perform the experiments in NG vs. PDM, and in PDM with vs. those without metformin.

Line 398: “ex-vivo” should be in italics.

We corrected it.

Line 408: “Thus, it results in the modulation of SIRT6 expression and inflammatory burden”. Incorrect English, please rephrase this sentence.

As suggested, we re-wrote this sentence in a correct English.

(x) English language and style are fine/minor spell check required

As suggested, we revised the English language and style (fine/minor spell check) by a native English mother tongue.